

# Assessing the risk of concurrent mycoplasma pneumoniae pneumonia in children with tracheobronchial tuberculosis: retrospective study

Lin Liu[1,*], Jie Jiang[1,*], Lei Wu[1], De miao Zeng[2], Can Yan[1], Linlong Liang[1], Jiayun Shi[1] and Qifang Xie[1]

[1] Department of Pediatrics, the Affiliated Changsha Central Hospital, Hengyang Medical School, University of South China, Changsha, Hunan, China
[2] Department of Joint Surgery, he Hong-he Affiliated Hospital of Kunming Medical University/ The Southern Central Hospital of Yun-nan Province (The First People's Hospital of Honghe State), Changsha, Hunan, China
* These authors contributed equally to this work.

Corresponding author
Qifang Xie, fangon@163.com

## ABSTRACT

**Objective:** This study aimed to create a predictive model based on machine learning to identify the risk for tracheobronchial tuberculosis (TBTB) occurring alongside *Mycoplasma pneumoniae* pneumonia in pediatric patients.

**Methods:** Clinical data from 212 pediatric patients were examined in this retrospective analysis. This cohort included 42 individuals diagnosed with TBTB and *Mycoplasma pneumoniae* pneumonia (combined group) and 170 patients diagnosed with lobar pneumonia alone (pneumonia group). Three predictive models, namely XGBoost, decision tree, and logistic regression, were constructed, and their performances were assessed using the receiver's operating characteristic (ROC) curve, precision-recall curve (PR), and decision curve analysis (DCA). The dataset was divided into a 7:3 ratio to test the first and second groups, utilizing them to validate the XGBoost model and to construct the nomogram model.

**Results:** The XGBoost highlighted eight significant signatures, while the decision tree and logistic regression models identified six and five signatures, respectively. The ROC analysis revealed an area under the curve (AUC) of 0.996 for XGBoost, significantly outperforming the other models ($p < 0.05$). Similarly, the PR curve demonstrated the superior predictive capability of XGBoost. DCA further confirmed that XGBoost offered the highest AIC (43.226), the highest average net benefit (0.764), and the best model fit. Validation efforts confirmed the robustness of the findings, with the validation groups 1 and 2 showing ROC and PR curves with AUC of 0.997, indicating a high net benefit. The nomogram model was shown to possess significant clinical value.

**Conclusion:** Compared to machine learning approaches, the XGBoost model demonstrated superior predictive efficacy in identifying pediatric patients at risk of concurrent TBTB and *Mycoplasma pneumoniae* pneumonia. The model's identification of critical signatures provides valuable insights into the pathogenesis of these conditions.

# INTRODUCTION

Tuberculosis (TB), recognized as a Category B notifiable infectious disease in China, is preventable and manageable (*MacNeil et al., 2020*). Annually, TB affects over 2 million children globally, leading to the death of over 200 children every day (*Shakoor & Mir, 2022*). The non-specific manifestations of pediatric TB frequently lead to its misdiagnosis or underdiagnosis, suggesting that its actual prevalence could surpass current estimates (*Marangu & Zar, 2019*; *van Heerden et al., 2021*). Despite a reported decrease in TB incidence within China, the country's large population presents a persistent, significant ongoing health challenge. Consequently, China remains among the nations with the highest TB burden globally (*Pang et al., 2019*).

Tracheobronchial tuberculosis (TBTB) explicitly targets the mucosa, submucosa, smooth muscle, cartilage, and epithelium of the trachea and bronchi, classifying it as a distinct TB subtype that affects the lower respiratory tract (*Liu et al., 2022*). Pediatric patients with TBTB present with non-specific symptoms, such as fever, cough, and wheezing. However, these can lead to critical complications, such as airway constriction, occlusion, sensitivity, and pulmonary atelectasis (*Zhuang et al., 2023*), highlighting the need for early diagnosis and prompt intervention to enhance patient prognosis and quality of life.

Conversely, lobar pneumonia, primarily caused by *Streptococcus pneumoniae*, is more frequently observed in young adults. This condition is characterized by the abrupt onset of high fever, chills, chest discomfort, cough, and dyspnea (*Zhuang et al., 2023*). The primary pathology initiates in the alveoli and rapidly spreads to affect one or several lobes. *Mycoplasma pneumoniae* pneumonia constitutes 10% to 40% of pediatric hospitalizations, exhibiting cyclical outbreaks every 3 to 7 years. Each outbreak can persist for a year, marking significant periodic public health challenges (*Rothberg, 2022*).

Distinguishing between TBTB and lobar pneumonia presents a diagnostic challenge, as both conditions frequently start with non-specific symptoms such as fever, cough, and dyspnea (*Kim et al., 2020*). From a radiological perspective, both diseases show signs of inflammatory lung changes, including solid masses or pulmonary infiltrates.

In light of the symptomatic overlap between TBTB and lobar pneumonia, this study introduces an innovative machine learning-based risk prediction model to enhance diagnostic precision. This model, particularly through the XGBoost algorithm, represents a significant advancement in clinical diagnostics, substantially reducing the risk of misdiagnosis. It provides a sophisticated tool for clinicians to differentiate between these conditions more accurately, ensuring targeted and effective patient treatment. Furthermore, this approach enhances diagnostic accuracy and contributes to a deeper understanding of the pathophysiology of these diseases, thereby establishing a new benchmark in their management.

## MATERIALS AND METHODS

### Clinical registration

This study was registered with the China Clinical Trial Registry, designated by the registration number ChiCTR2300076648.

### Ethical approval

The Medical Ethics Committee of Changsha Central Hospital in Hunan Province granted ethical clearance for this investigation (Ethics Approval No.: 2023-057).

### Sample size determination

Based on the reported incidence rate of pediatric TBTB alongside *Mycoplasma pneumoniae* pneumonia at 5% (*Peng et al., 2020*; *Wang et al., 2020*), the calculation for sample size was determined using the formula: $n = \frac{Z^2 \times p \times (1-p)}{E^2}$. In this formula, "$n$" is the sample size needed, "Z" is the z-value corresponding to a specified level of confidence (at $\alpha = 0.05$, Z = 1.96 for a two-sided test), "$p$" is the estimated incidence rate (5% or 0.05), and "E" is the margin of error (set at 0.05). This calculation yielded a required sample size of approximately 384 cases. The sample size was adjusted to 428 cases to account for a potential 10% dropout rate. This sample size might be recalibrated depending on the study's interim findings.

### Sample source

This study included 320 pediatric patients treated for *Mycoplasma pneumoniae* pneumonia between June 1, 2018, and January 1, 2023.

### Inclusion and exclusion criteria
#### Inclusion criteria

1) Patients with a clinical diagnosis of pediatric *Mycoplasma pneumoniae* pneumonia (*Huang et al., 2021*).
2) Those meeting the diagnostic criteria for either TBTB or lobar pneumonia.
3) Those who underwent standard diagnostic procedures.
4) Availability of comprehensive clinical records.
5) Children presenting for their initial consultation without prior treatment for the conditions under study.

#### Exclusion criteria

1) Patients with severe cardiac, hepatic, or renal dysfunction.
2) Individuals with concurrent pulmonary diseases.
3) Cases of autoimmune diseases.
4) Presence of circulatory system disorders.
5) Diagnosis of any malignant tumors.
6) Infections caused by pathogenic microorganisms other than the focus of this study.

### Diagnostic criteria

*For TBTB*

Diagnosis of TBTB was established by the People's Republic of China's health industry standards "Diagnosis of Pulmonary TB (WS288-2017; *Administration Office, Beijing Children's Hospital, 2018*)." A confirmed diagnosis required direct visualization of tracheal and bronchial lesions *via* fiber-optic bronchoscopy. Furthermore, a positive result from any of the following: biopsy pathology, *Mycobacterium tuberculosis* smears from secretions, cultures, or nucleic acid testing, was necessary for diagnosis (*Vonasek et al., 2021*).

*For lobar pneumonia*

Diagnosis follows criteria from "Zhufutang Practical Pediatrics" (*Administration Office, Beijing Children's Hospital, 2018*), including clinical symptoms of cough, fever, and dyspnea, diminished respiratory sounds with dry or wet rales at the affected site, and computerized tomography (CT) scans showing inflammatory infiltrative lesions or segmental broad, dense shadows in the lung parenchyma.

## Sample screening

Out of the initial cohort, 212 samples fulfilled the inclusion, exclusion, and diagnostic criteria. Of these, 170 cases identified with lobar pneumonia formed the "pneumonia group," and 42 cases diagnosed with TBTB alongside *Mycoplasma pneumoniae* pneumonia constituted the "combined group." A flowchart was constructed to detail the sample selection process (Fig. 1).

## Clinical data collection

Clinical data for pediatric patients were collected from electronic medical records and follow-up outpatient records. Data encompassed age, Gender, geographic region, history of fever, presence and duration of cough, presence of wet rhonchi, and counts of white blood cells (WBC), neutrophils (NEU), lymphocytes (LYM), hemoglobin (HGB), platelets (PLT), monocytes (MONO), C-reactive protein (CRP), lactate dehydrogenase (LDH), neutrophil-to-lymphocyte ratio (NLR), platelet-to-lymphocyte ratio (PLR), monocyte-to-lymphocyte ratio (MLR), the Systemic Immune-Inflammation Index (SII), and the Systemic Immune-Inflammation Index-Related Index (SIRI).

## Peripheral blood ratio calculation

NLR = Neutrophil count/Lymphocyte count.
PLR = Platelet count/Lymphocyte count.
MLR = Monocyte count/Lymphocyte count.
SII = Platelet count × (Neutrophil count/Lymphocyte count).
SIRI = (Neutrophil count × Monocyte count)/Lymphocyte count.

## Blood analysis

Upon the admission of patients, blood samples were analyzed using the Xisenmechem XT-1800i analyzer to measure routine blood indices.

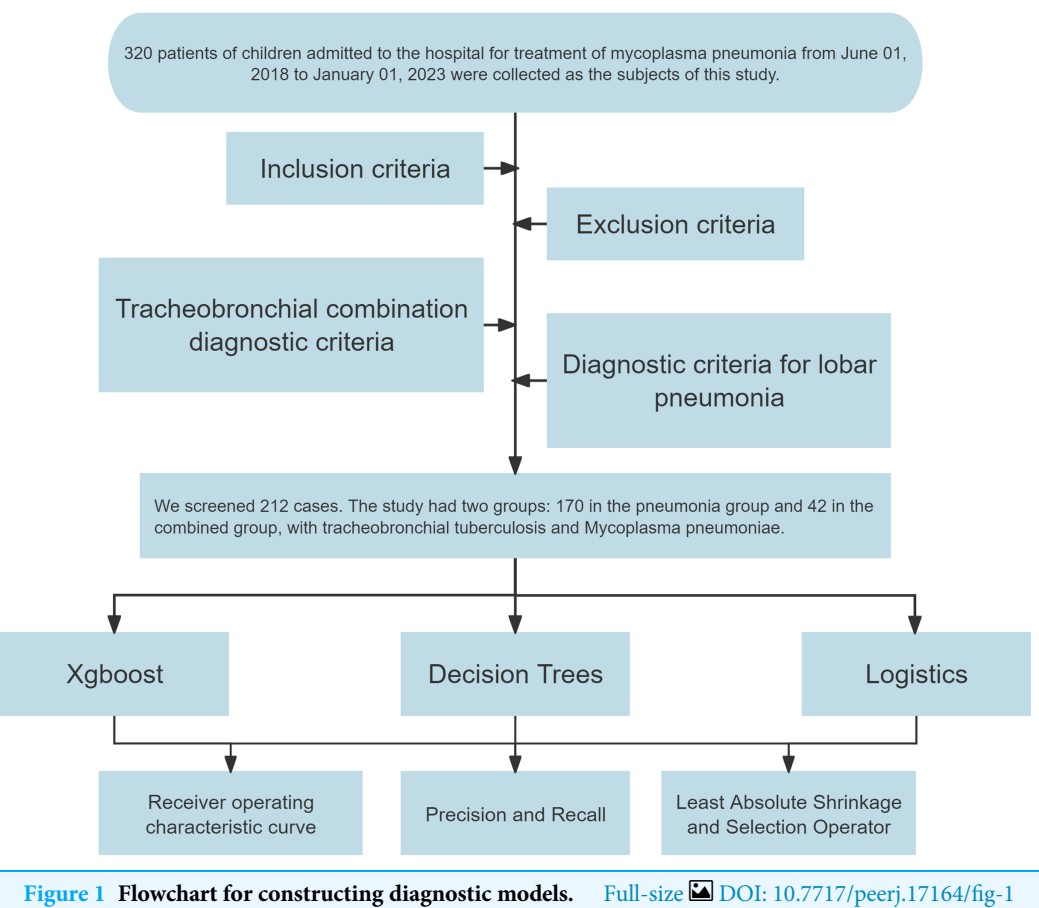

**Figure 1 Flowchart for constructing diagnostic models.**

## XGBoost model construction

The XGBoost algorithm was harnessed for a binary classification analysis of the clinical information dataset. The "Subgroups" column was encoded as the target label for the model. The XGBoost framework utilized "gbtree" as its booster, employing binary logistic regression for its objective function and using the area under the curve (AUC) metric for model evaluation. Key parameters were a maximum tree depth of six, a learning rate set of 0.01, and sample and column subsampling rates at 0.8. The training was halted after ten iterations without performance gains. The model's efficacy was gauged post-training *via* receiver's operating characteristic (ROC) curves and the determined AUC values. Furthermore, the relevance of different data signatures was determined and depicted visually (*Gan, 2022*).

## Decision tree

The "rpart" package in R was used for decision tree analysis, setting specific parameters. The "rpart" function was employed for classification tasks with the formula "Subgroups~.," which included all features except the target label "Subgroups" for the model training. The method parameter was set to "class" to indicate the model's focus on classification. Model predictions were obtained using the "predict" function, with the type set to "prob," facilitating the extraction of probabilities for the positive class. The "roc" function from the

"pROC" package, along with the "auc" function, were utilized to plot the ROC curve and calculate the AUC value, which is critical for evaluating the model's predictive performance. The "importance" attribute from the output of the "rpart" model was employed to assess and depict the significance of features, with the calculation of importance percentages emphasizing the most influential variables. This comprehensive analytical strategy highlights the meticulousness and precision of our statistical methodology (*Yu & Wang, 2022*).

### Logistic regression

Logistic regression provides a linear modeling technique for binary or multiclass classification tasks. Unlike linear regression, which predicts a continuous output, logistic regression estimates a probability representing the likelihood of an event's occurrence based on input attributes. This model employs a "sigmoid" or "softmax" function (in multiclass scenarios) to map predicted values to the [0,1] interval, which is then classified according to this derived probability (*Wickett & Carrasco, 2011*).

### Outcome measures

(1) The characteristics of pediatric patients with concurrent TBTB and *Mycoplasma pneumoniae* pneumonia identified using XGBoost, decision tree, and logistics regression models. (2) The predictive performance of the three models for pediatric TBTB combined with *Mycoplasma pneumoniae* pneumonia. (3) The predictive accuracies of the three models in diagnosing children with TBTB combined with *Mycoplasma pneumoniae* pneumonia were compared using the DeLong test. (4) The clinical applicability of the XGBoost model was assessed by distributing 212 specimens into two validation groups, using a partition ratio of 7:3.

### Statistical analysis

Data preprocessing was conducted using SPSS software (version 26.0). R (version 4.2.2; *R Core Team, 2023*) software was utilized for the subsequent statistical analysis.
The "xgboost" package in R facilitated the application of the XGBoost algorithm, providing an effective, efficient, and scalable approach to predictive modeling. The construction, assessment, and visualization of complex decision trees were made possible through the "rpart" package. Decision Curve Analysis (DCA) and the generation of ROC curves were performed using "media" and "rock" packages, respectively. The "rms" package was instrumental in creating nomograms, enhancing the interpretability of our results.
The DeLong test was utilized to evaluate the predictive accuracies of the three models in diagnosing pediatric patients with concurrent TBTB and *Mycoplasma pneumoniae* pneumonia. Statistical significance was set at $p < 0.05$.

## RESULTS

### Signature identification using XGBoost

The XGBoost algorithm was applied to the patient data to identify significant variable associations (Fig. 2A). Subsequently, those with <1% contribution were excluded, leaving

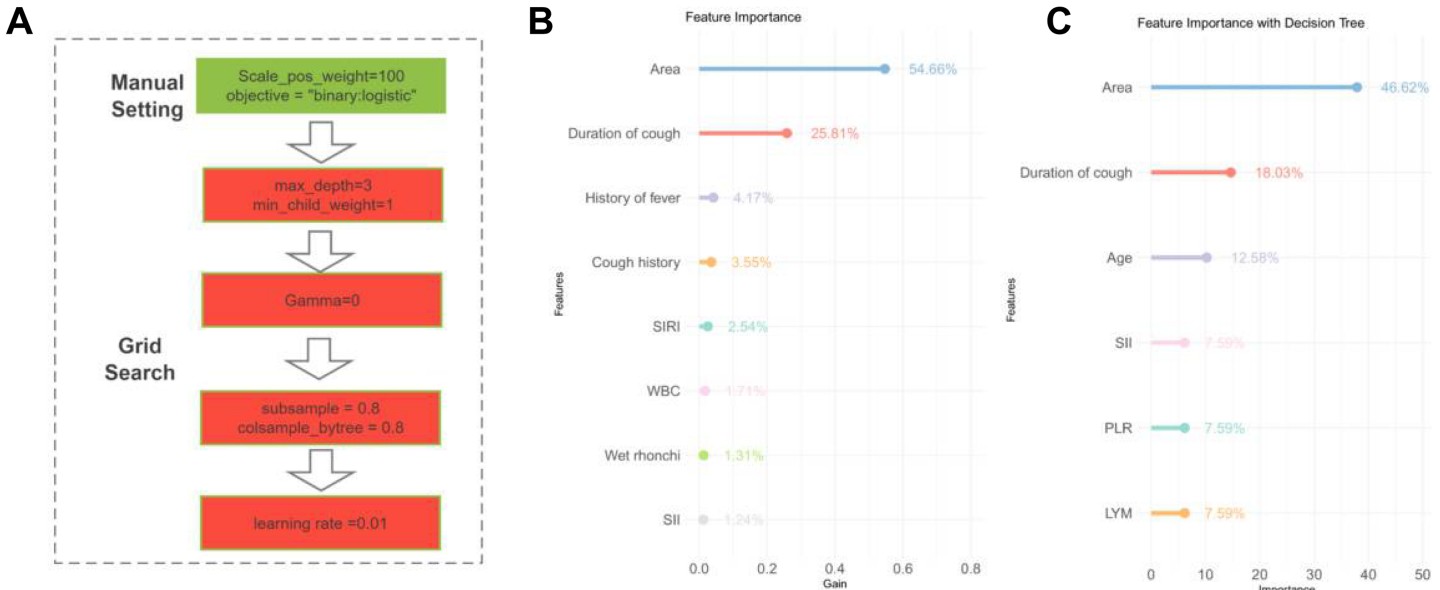

**Figure 2 Signature variable selection using XGBoost and decision tree.** (A) Workflow diagram for XGBoost methodology. (B) XGBoost-based signature importance for identifying characteristic variables in children with tracheobronchial tuberculosis combined with *Mycoplasma pneumoniae* pneumonia. (C) Decision tree analysis highlighting key variables for diagnosing tracheobronchial tuberculosis combined with *Mycoplasma pneumoniae* pneumonia in children. Note: Abbreviations: WBC, white blood cells; LYM, lymphocytes; PLR, platelet-to-lymphocyte ratio; SII, Systemic Immune-Inflammation Index; SIRI, Systemic Immune-Inflammation Index-Related Index.

us with a refined list of eight critical signatures: geographical area, duration of cough, history of fever, cough history, SIRI, WBC count, presence of wet rhonchi, and SII.

## Signature selection through decision tree

The decision tree model was executed on the dataset to discern the importance of various variables (Fig. 2B). Variables with a weight of <1% were considered negligible and omitted. This filtration led to the identification of six key signatures: geographic region, duration of cough (in days), patient age, PLR, SII, and LYM.

## Signature selection through logistic regression

Signature selection was analyzed using logistic regression. Univariate analysis highlighted significant differences between the two groups in various variables, including age, geographic region, history of fever, cough history, duration of cough, presence of wet rhonchi, and counts of WBC, NEU, LYM, PLT, MONO, LDH, and ratios of NLR, PLR, MLR, SII, and SIRI ($p < 0.05$, Tables 1 and 2). These variables were binarized by establishing cut-off values through ROC curve analysis (Table 2). Subsequent multifactorial logistic regression analysis revealed that geographical region, fever history, cough history, duration of cough, and WBC levels were significant risk factors for pediatric patients with concurrent TBTB and *Mycoplasma pneumoniae* pneumonia ($p < 0.05$, Table 3).

**Table 1  Univariate analysis of factors using logistic regression.**

| Factors | Combined group ($n$ = 42) | Pneumonia group ($n$ = 170) | Z/$\chi^2$ | $p$ |
|---|---|---|---|---|
| Age | 11.5 (4.25, 15) | 7 (5.00, 8.00) | 3.063 | 0.002 |
| Gender (M/F) | 16/26 | 84/86 | 1.731 | 0.188 |
| Location Type (urban/rural) | 8/34 | 159/11 | 111.739 | <0.001 |
| History of fever (yes/no) | 19.23 | 154/16 | 46.140 | <0.001 |
| History of cough (yes/no) | 36/6 | 169/1 | 5.611 | <0.001 |
| Duration of cough (days) | 26 (10.00, 57.50) | 4 (3.00, 6.00) | 6.341 | <0.001 |
| Wet rhonchi (yes/no) | 10/32 | 83/87 | 8.557 | 0.003 |

**Table 2  Laboratory findings.**

| Factors | Combined group ($n$ = 42) | Pneumonia group ($n$ = 170) | Z/t/$\chi^2$ | $p$ |
|---|---|---|---|---|
| WBC ($10^9$/mL) | 9.3 (6.77, 10.87) | 6.715 (5.49, 8.42) | 4.322 | <0.001 |
| NEU ($10^9$/mL) | 66.09 ± 14.09 | 61.55 (55.60, 68.40) | 2.600 | 0.009 |
| LYM ($10^9$/mL) | 24.55 ± 14.02 | 29.38 ± 9.35 | −2.122 | 0.039 |
| HGB (g/dL) | 118.45 ± 19.01 | 123 (116.00, 128.00) | −0.931 | 0.381 |
| PLT ($10^9$/mL) | 344.98 ± 114.68 | 249 (206.00, 296.00) | 4.617 | <0.001 |
| MONO ($10^9$/mL) | 0.74 (0.54, 0.87) | 0.54 (0.44, 0.67) | 4.353 | <0.001 |
| CRP (mg/L) | 18.25 (7.55, 38.22) | 14.44 (6.15, 26.82) | 1.226 | 0.175 |
| LDH (U/L) | 265.50 (217.00, 325.50) | 318.00 (266.00, 375.00) | −3.726 | <0.001 |
| NLR | 2.95 (1.68, 4.83) | 2.14 (1.53, 3.01) | 2.890 | 0.004 |
| PLR | 13.75 (8.72, 28.57) | 8.616 (6.36, 11.83) | 4.598 | <0.001 |
| MLR | 0.03 (0.01, 0.06) | 0.01 (0.01, 0.02) | 4.084 | <0.001 |
| SII | 878.86 (539.51, 2,165.14) | 551.26 (371.77, 752.00) | 4.230 | <0.001 |
| SIRI | 2.19 (1.03, 5.28) | 1.16 (0.781, 1.858) | 3.846 | <0.001 |

Note:
WBC, white blood cells; NEU, Neutrophils; LYM, lymphocytes; HGB, hemoglobin; PLT, platelets; MONO, monocytes; CRP, C-reactive protein; LDH, Lactate dehydrogenase; NLR, neutrophil-to-lymphocyte ratio; PLR, platelet-to-lymphocyte ratio; MLR, monocyte-to-lymphocyte ratio; SII, Systemic Immune-Inflammation Index and SIRI, Systemic Immune-Inflammation Index-Related Index.

**Table 3  Multivariable logistic regression analysis results.**

| Factors | B | SD | Chi-square | $p$ | OR | 95% CI Low | 95% CI High |
|---|---|---|---|---|---|---|---|
| Area | −6.593 | 1.639 | 16.181 | <0.001 | 0.001 | 0.000 | 0.034 |
| History of fever | −2.239 | 1.039 | 4.644 | 0.031 | 0.107 | 0.014 | 0.816 |
| History of cough | −7.922 | 3.701 | 4.581 | 0.032 | <0.001 | 0.000 | 0.513 |
| Coughing time | 6.162 | 1.645 | 14.029 | <0.001 | 474.508 | 18.871 | 11,931.368 |
| WBC | 3.576 | 1.248 | 8.211 | 0.004 | 35.715 | 3.095 | 412.071 |

Note:
WBC, white blood cells; B, coefficient estimate; SD, standard deviation; OR, odds ratio; CI, confidence interval.

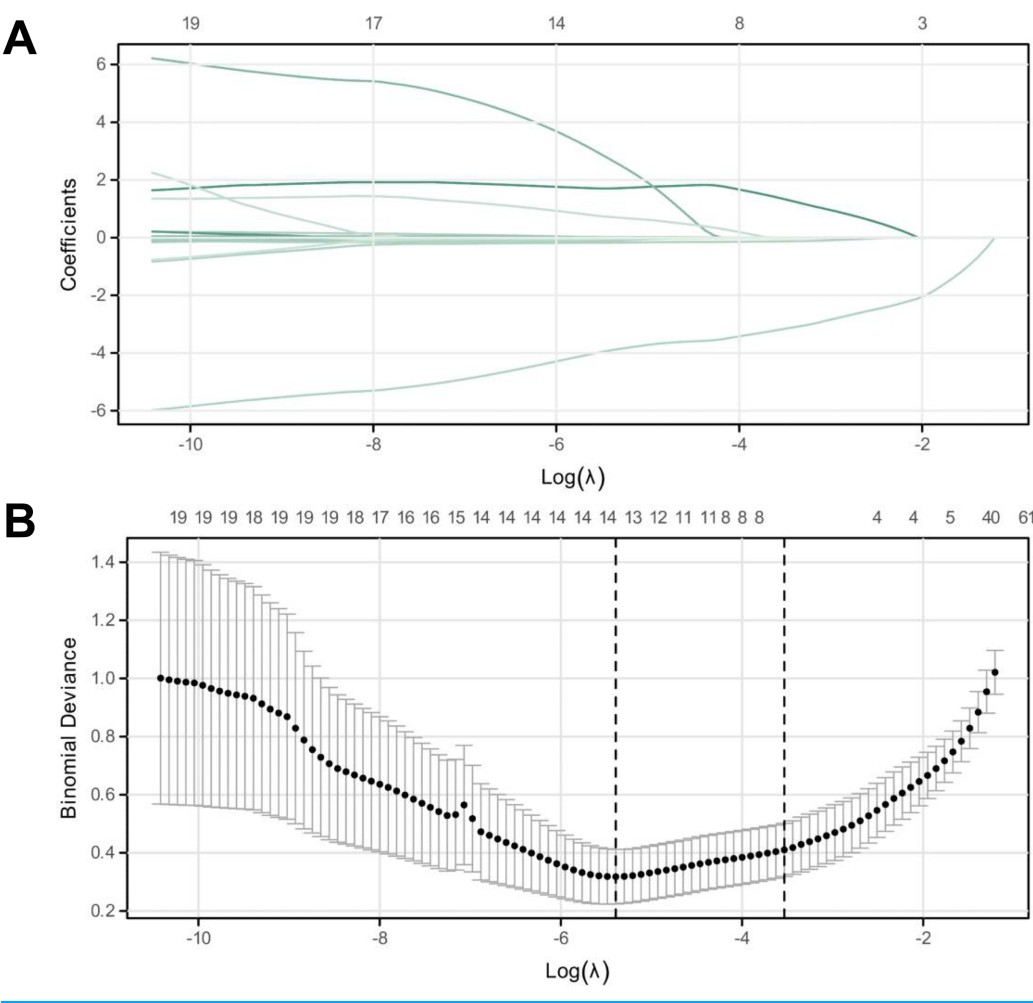

**Figure 3 LASSO regression analysis for identifying characteristic variables.** (A) The cross-validation fit (cvfit) for model selection using LASSO regression. (B) Lambda curve, illustrating the regularization path of LASSO regression. The model was performed based on the minimum criteria.

**Table 4 Detailed parameters of the ROC curve.**

| Variables | AUC | CI | Sensitivity | Specificity | Jordon Index |
|---|---|---|---|---|---|
| XGBoost | 0.996 | [0.992–1.000] | 97.64% | 97.61% | 95.26% |
| Decision tree | 0.943 | [0.905–0.982] | 91.77% | 95.24% | 87.00% |
| Logistics | 0.851 | [0.761–0.941] | 87.65% | 83.33% | 70.98% |

## Evaluation of model performance

Upon concluding the study, each model's performance was compared by examining the ROC and PR prediction curves. The AUC values were as follows: XGBoost: 0.992, decision tree: 0.943, and logistic regression: 0.851 (Figs. 3A and 3B, Table 4). The DeLong test affirmed the XGBoost model's superior predictive capabilities over the decision tree and logistic regression models ($p < 0.01$, Table 5), ranking them as XGBoost > decision tree >

**Table 5 DeLong test comparing the effectiveness of the three models.**

|  | z | p | Difference between AUC | Difference between SD | 95% CI | |
|---|---|---|---|---|---|---|
|  |  |  |  |  | Low | High |
| XGBoost-decision tree | 2.787 | 0.005 | 0.053 | 0.148 | 0.016 | 0.091 |
| XGBoost-logistics | 3.156 | 0.002 | 0.145 | 0.219 | 0.055 | 0.236 |
| Decision trees-logistics | 2.019 | 0.044 | 0.092 | 0.255 | 0.003 | 0.181 |

**Note:**
z-score, test statistic in DeLong test; SD, standard deviation; AUC, area under the curve; CI, confidence interval.

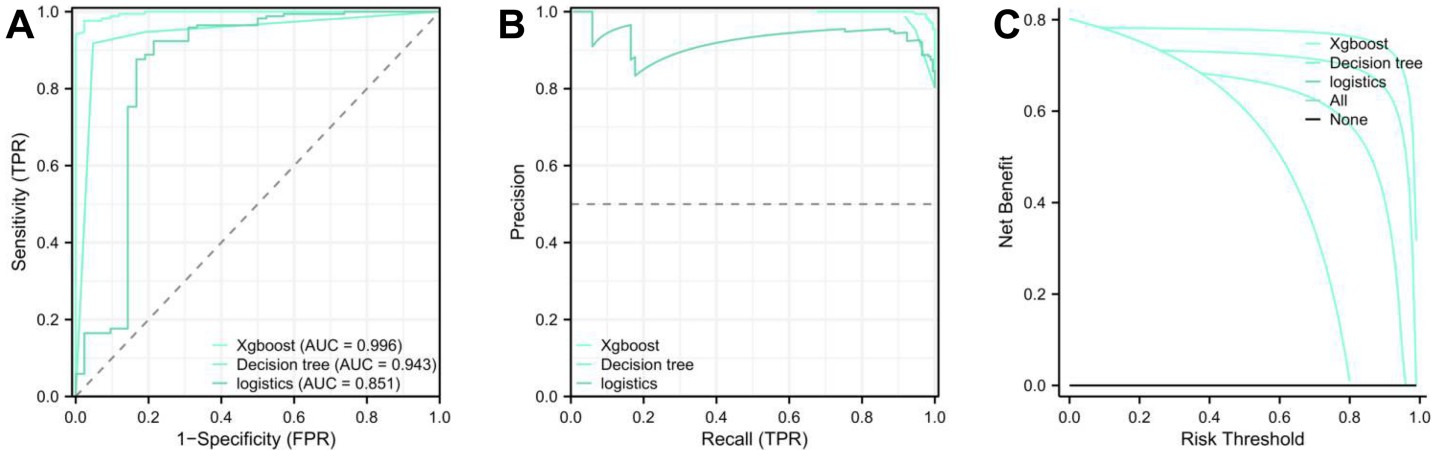

**Figure 4 Validation and nomogram construction of XGBoost model.** (A) ROC, PR, and DCA analyses for the first validation group of the XGBoost model, assessing energy efficiency and clinical relevance. (B) ROC, PR, and DCA analyses for the second validation group of the XGBoost model, evaluating clinical relevance. (C) Nomogram visualization for the XGBoost model based on eight key factors.

logistic regression. Furthermore, the Akaike Information Criterion (AIC) assessment revealed the XGBoost model (AIC = 43.226) had a better fit compared to the Decision Tree (87.259) and logistic regression models (135.231) (Fig. 3C). The XGBoost model demonstrated the highest average net benefit of 0.764227, outperforming the decision tree (0.708071) and logistic regression models (0.625883). Consequently, the XGBoost model is recommended for diagnosing pediatric patients with TBTB coexisting with *Mycoplasma pneumoniae* pneumonia (Table S1).

## XGBoost model validation

The original dataset was divided into validation groups 1 and 2, following a 7:3 distribution to validate the XGBoost model. Analysis of the ROC and PR curves revealed an AUC of 0.997 for both groups. The AIC values were 21.571 for the validation one group and 13.612 for the validation teo group, indicating a robust model fit. The average net benefits within the XGBoost model were 0.804 for validation 1 and 0.796 for validation two, demonstrating the model's predictive accuracy (Figs. 4A and 4B). Furthermore, a nomogram was constructed based on eight characteristic factors. Using the profile of a hypothetical pediatric patient (city, fever history, cough history, cough duration of 15 d, absence of wet rhonchi, WBC count of 15.99, SII of 3,479.47, and SIRI of 9.01), a morbidity

prediction score of (0 + 40 + 100 + 76 + 0 + 26 + 30 + 15) 287 points, corresponding to a 72% morbidity rate for this child (Fig. 4C).

## DISCUSSION

*Mycoplasma pneumoniae* pneumonia represents a significant respiratory infection in children, characterized by symptoms such as cough and fever, potentially leading to long-term pulmonary damage (*Wang et al., 2023*; *Zhu et al., 2023*). Although imaging diagnostics are crucial, their frequent application in pediatric care raises concerns about elevated radiation exposure risks (*Kersemans et al., 2011*), highlighting the importance of weighing the diagnostic advantages against potential health risks.

Emerging technological innovations have marked the beginning of a new era in diagnostics through machine learning (*Nature Medicine, 2023*; *Nature, 2023*). Machine learning empowers physicians to expedite and refine diagnostic accuracy by analyzing extensive biomedical datasets. Algorithms in machine learning pave the way for the development of automated and unbiased diagnostic models, potentially diminishing the reliance on imaging modalities and subsequently decreasing the radiation exposure risk for children (*Garriga et al., 2022*). Complex, high-dimensional biomedical data could be comprehensively analyzed through the proactive adoption of these algorithms. Such analysis improves diagnostic precision and offers critical insights into monitoring therapeutic efficacy (*Swanson et al., 2023*). As an invaluable tool to truncate diagnostic latency, machine learning provides significant potential for the early detection and intervention of diseases, potentially saving numerous lives.

The landscape of modern medical research and clinical diagnostics has shifted considerably from traditional paradigms to the integration of data analytics and machine learning. Within this evolving field, models such as XGBoost, decision tree, and logistic regression have gained prominence, showcasing their pivotal significance in clinical medicine (*Kalyani et al., 2023*). This study aimed to ascertain the diagnostic efficacy of these models in pediatric patients with coexisting TBTB and *Mycoplasma pneumoniae* pneumonia.

Based on ROC and PR curves, our assessment demonstrated that the XGBoost model, with an AUC value of 0.992, surpassed its counterparts in diagnosing this specific pediatric condition. The DeLong test further accentuated XGBoost's diagnostic superiority. The effectiveness ranking was XGBoost > decision tree > logistic regression. The AIC metric endorsed XGBoost's superior model fit with a score of 43.226. Furthermore, DCA indicated XGBoost's commendable average net benefit of 0.764227, the highest among all evaluated models. These findings suggest XGBoost as the preferred choice for diagnosing children with concurrent *Mycoplasma pneumoniae* pneumonia and TBTB, promising superior diagnostic accuracy and reliability.

The variability in the performance of different models may be ascribed to factors such as the models' complexity, their ability to extract relevant signatures, control overfitting, the quality and volume of data, and the tuning of parameters. XGBoost stands out for its adaptability and potent predictive capabilities when handling complex clinical datasets. Although decision tree models exhibit reasonable efficacy, they potentially lag behind

XGBoost in certain aspects due to their inherent simplicity and absence of a sophisticated signature selection mechanism, making logistic regression models less suitable for complex diagnostic challenges due to their foundational simplicity and lack of a robust signature selection process.

Supporting these findings, *Hu et al. (2022)* demonstrated that XGBoost outperformed LASSO in internal validation, though LASSO showed a marginal advantage in external validation. Furthermore, XGBoost was superior to logistic regression models in predicting outcomes for hepatocellular and gallbladder carcinomas (*Zhang et al., 2021*).

Selecting the appropriate machine learning model selection is paramount in clinical diagnostics and prognostics. For instance, a study on blood pressure variation during hemodialysis analyzed various models without determining a definitive best-performing model (*Huang et al., 2020*), suggesting that the model's efficacy may depend on specific dataset characteristics and task specifications. Similarly, research involving patients with COVID-19 assessed the effectiveness of multiple models without furnishing comparative outcomes, indicating that model efficacies could vary across distinct disease domains and objectives (*Hong et al., 2022*). Another comparative study between XGBoost and logistic regression for predicting ICU mortality in patients with rheumatic heart disease underscored the importance of comparing machine learning models to enhance clinical diagnostics (*Xu et al., 2022*). Therefore, selecting an appropriate model that aligns with the specific clinical challenge and dataset is crucial (*Adamson & Welch, 2019*).

Our investigation identified eight significant variables intrinsically linked to pediatric TBTB coexistent with *Mycoplasma pneumoniae* pneumonia utilizing the XGBoost model. The interplay of these variables aligns logically within biological and clinical science realms.

Factors such as limited healthcare infrastructure and inadequate sanitation in rural settings may predispose children to heightened exposure to *Mycobacterium tuberculosis* and other respiratory pathogens (*Pinto et al., 2023*). An extended duration of cough may indicate the persistent and severe nature of these respiratory infections. A prolonged cough, without a history of fever, could serve as an early indicator of TBTB, while fewer cough episodes might suggest heightened sensitivity or susceptibility to respiratory pathogens among children (*Kim et al., 2022*). The SIRI and leukocyte counts potentially reflect the body's immunological response to infection, signaling acute inflammatory reactions. A decrease in the incidence of wet rales suggests obstructions in respiratory mucus due to pathogenic activity, aligning with the pathogenesis of *Mycoplasma pneumoniae* pneumonia (*Venkatappa et al., 2023*). An increased SII supports the presence of significant inflammation and infection, corroborating the severity of the concurrent conditions (*VanValkenburg et al., 2022*). Analyzing these variables and their association with the disease provides a deeper understanding of the pathophysiology and clinical presentation of TBTB in conjunction with *Mycoplasma pneumoniae* pneumonia. However, supplemental research endeavors are warranted to elucidate the complex interactions among these variables and their impact on the disease and to probe the underlying biological processes.

A validation phase was incorporated for the XGBoost model into our study for further introspection. This process involved dividing the sample into two validation groups, 1 and 2, using a 7:3 ratio. The outcomes showcased exceptionally high AUC values of 0.997 for both groups, confirming the model's high predictive accuracy and reliability across varying datasets. The AIC scores and the calculations of average net benefit further validated the model's efficacy. Furthermore, a nomogram constructed from eight characteristic variables demonstrated its practical application by estimating a 72% morbidity rate for a hypothetical patient case. This extension of our initial analysis emphasizes the potential of machine learning in enhancing diagnostic precision, underscoring the significance of our methodological contributions to the discipline.

The study acknowledges several limitations that merit consideration. The relatively small sample size may not have fully represented the diversity of the target population. The retrospective design of the study may have introduced selection bias. Although we evaluated three machine learning models, numerous other models could apply to clinical diagnostics. Furthermore, excluding certain risk factors and possible confounders could have influenced the study's results and interpretations. Expanding the sample size, employing a prospective design, exploring a broader selection of models, and encompassing a comprehensive set of risk factors are recommended for enhanced precision and reliability.

## CONCLUSIONS

This study demonstrates the exceptional diagnostic precision of the XGBoost model in identifying cases of pediatric TBTB with *Mycoplasma pneumoniae* pneumonia and identifying eight critical factors that offer insights into the disease's pathogenesis. The observation highlights the significant potential of machine learning technologies, particularly the XGBoost model, in refining clinical diagnostic processes and their prospective incorporation into medical practice.

### Funding
The authors received no funding for this work.

### Competing Interests
The authors declare that they have no competing interests.

### Author Contributions
- Lin Liu conceived and designed the experiments, analyzed the data, prepared figures and/or tables, and approved the final draft.
- Jie Jiang conceived and designed the experiments, analyzed the data, prepared figures and/or tables, and approved the final draft.
- Lei Wu performed the experiments, prepared figures and/or tables, and approved the final draft.

- De miao Zeng performed the experiments, authored or reviewed drafts of the article, and approved the final draft.
- Can Yan performed the experiments, authored or reviewed drafts of the article, and approved the final draft.
- Linlong Liang performed the experiments, authored or reviewed drafts of the article, and approved the final draft.
- Jiayun Shi performed the experiments, authored or reviewed drafts of the article, and approved the final draft.
- Qifang Xie analyzed the data, prepared figures and/or tables, and approved the final draft.

## Human Ethics

The following information was supplied relating to ethical approvals (*i.e.*, approving body and any reference numbers):

The study was approved by the Ethics Committee Changsha Central Hospital, ethical approval number: 2023-057. And all studies were conducted in accordance with the relevant guidelines, following the Declaration of Helsinki, and informed consent was obtained from all participants or/and their legal guardians. It has been in in Chinese Clinical Trial Register), clinical trial registration number: ChiCTR2300076648.

## Data Availability

The data from this study are available at Zenodo: Qifang, X. (2023). Assessing the Risk of Concurrent *Mycoplasma pneumoniae* Pneumonia in Children with Tracheobronchial Tuberculosis. https://doi.org/10.5281/zenodo.10434187.

## Clinical Trial Registration

The following information was supplied regarding Clinical Trial registration:

ChiCTR2300076648.

## Supplemental Information

Supplemental information for this article can be found online at http://dx.doi.org/10.7717/peerj.17164#supplemental-information.

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
