# Peer review of "Assessing the risk of concurrent mycoplasma pneumoniae pneumonia in children with tracheobronchial tuberculosis: retrospective study"

_PeerJ, doi:10.7717/peerj.17164_

## Round 0.1 · original submission · Major Revisions

Your manuscript presents valuable insights and employs advanced machine learning techniques in a significant area of pediatric healthcare. However, to enhance the clarity, credibility, and overall quality of the paper, these points need to be thoroughly addressed:

1. The manuscript requires careful proofreading to enhance the flow of language and ensure logical coherence between paragraphs. Please address the noted logical leaps or contradictory statements.

2. More detailed information on the source of the 5% incidence data, including references, is needed for clarity and validation.

3. A clear distinction between clinical and laboratory data is advisable.

4. Ensure a clearer statement of novelty in the introduction. Correct inconsistencies in data presentation, such as in Table 1. Consistency in the number of decimal places in data tables is also required. Simplify the conclusion section to focus on the main research question directly.

5. Address grammatical errors throughout the manuscript and enhance the quality and clarity of images, particularly Figure 2.

6. Please simplify the description in Figure 1. Long sentences should be replaced by short phrases.

7. Please clarify the statement regarding treatment received by all children in the study and its impact on the study results.

8. Please provide more description of how the "XGboost" and "Decision Tree" performed. Authors did not mention what related packages were used, In addition, figure legends for the LASSO regression should be explained more. What lambda(λ) was determined?

**Language Note:** The review process has identified that the English language must be improved. PeerJ can provide language editing services - please contact us at copyediting@peerj.com for pricing (be sure to provide your manuscript number and title). Alternatively, you should make your own arrangements to improve the language quality and provide details in your response letter. – PeerJ Staff

Reviewer 1 ·

Basic reporting

1. The article as a whole has a good flow of language, but the writer needs to take care to ensure logical coherence between paragraphs. Currently, there are some logical leaps or contradictory statements in the article that need to be adjusted. Overall, the manuscript needs to be carefully touched up and proofread.
2. Regarding sample size calculations, authors need to provide more detailed information on the source of the data on 5% incidence, including possible references.
3. regarding the collection of clinical data, authors are advised to categorize the data more clearly, e.g., into clinical and laboratory data, to avoid redundancy and possible misinterpretation of the current description.
4. although this is a retrospective study, there is also a need to describe the instruments used to examine the data, which would help to increase the transparency and credibility of the study.
5. the current formatting of the references does not meet the requirements of the journal, therefore the authors need to adjust the formatting of the references in the manuscript accordingly to meet the publication standards.

Experimental design

Based on this article, we can see that it meets a high standard in several aspects. The article successfully developed a risk prediction model for tracheobronchial tuberculosis combined with Mycoplasma pneumoniae pneumonia in children using machine learning methods, especially the XGBoost model. By comparing different models, XGBoost demonstrated superior predictive results, and in addition, the research problem was well defined, helping to fill in the gaps in existing knowledge with important clinical implications. Despite the technical and methodological excellence of the study, there are still some issues that need to be adjusted.
1. the relatively small sample size mentioned in the article may affect the generalizability of the findings.
2. due to the retrospective design of the study, there may be a risk of selection bias.
3. although the authors collected a small sample, the overall sample could have been split into two groups to verify the validity of the model.
Overall, this article excels in applying advanced machine learning techniques, but has some limitations in terms of sample size and study design that require further revision by the authors.

Validity of the findings

1. this paper needs a clearer statement of its novelty in the introductory section to highlight the innovative aspects of the study.
2. Although the authors have provided comprehensive and reliable basic data, there are some flaws in the presentation of the data. For example, in Table 1, although the Z-value is mentioned, what is actually presented in the table seems to be the result of Mann-Whitney U-test, which needs to be corrected.
3. The values in the data tables, especially the number of reserved places after the decimal point, should be consistent throughout the text.
4. In the conclusion section, the presentation is slightly lengthy and the main idea is not focused enough. It is recommended that the authors simplify and improve the conclusion by giving a clear answer to the research question directly to improve the overall quality and readability of the paper.

Additional comments

This study provides useful insights for risk prediction of tracheobronchial tuberculosis combined with Mycoplasma pneumoniae pneumonia in children, but its practical application in clinical practice has not been explored in detail.
1. how the study could better assist physicians in applying these findings to clinical decision-making and how the model could be integrated into existing healthcare systems
2. there is a relative lack of discussion in the study about the interpretability and operationalization of the model, which is particularly important for clinicians from non-technical backgrounds
3. It is recommended that the authors construct a Nomogram based on the factors that characterize the xgboost model screen to make it clinically valuable.

Reviewer 2 ·

Basic reporting

This article makes a notable contribution to the field of risk prediction for tracheobronchial tuberculosis combined with Mycoplasma pneumoniae pneumonia in children. The authors used advanced machine learning methods, compared the effectiveness of different models, and demonstrated in-depth analysis and the ability to synthesize and apply the techniques.
However, there are some shortcomings in this article. a. Grammatical errors: It is pointed out that there are some grammatical problems in the article, and it is suggested that the authors should make careful proofreading and revisions to improve the quality and readability of the article. b. Image clarity: Especially for Figure 2, it is suggested that the authors should improve the quality of the image and increase the number of pixels to ensure that the readers can understand the content of the figure clearly. c. Inclusion criteria issues. With regard to the statement that "all children had received other treatments," it is recommended that the authors clarify whether or not all children included in the study had received treatment and how this affects the interpretation of the results of the study. d. In the statistical analysis, the authors used the Delong test, but we do not see a description of this test in the description of the methodology.

Experimental design

a.The research conducted in the article is indeed an original and preliminary study that is consistent with the purpose and scope of the journal. It introduces a new methodology to develop a risk prediction model for a specific pediatric disease, tracheobronchial tuberculosis combined with Mycoplasma pneumoniae pneumonia, using machine learning techniques. b.The article specifies the purpose of the study, which focuses on developing risk prediction models for children with tracheobronchial tuberculosis combined with Mycoplasma pneumoniae pneumonia. This research utilizes advanced machine learning methods to improve the diagnostic accuracy of this disease, thus effectively bridging the gap in existing knowledge. c.The paper provides the approval number for ethical review and also the approval number for clinical medical registration, which is very important. d.Regarding the description of the machine learning model, the authors could have organized them into one paragraph, which is a bit redundant at the current length. e.Checking the definitions of abbreviations throughout the text, we found that some of the tables and images in the manuscript are annotated with abbreviations underneath, while some are not annotated underneath.

Validity of the findings

a.This article is truly original in its research methodology. It addresses a well-defined and relevant research question by using a machine learning approach to develop a risk prediction model for children with tracheobronchial tuberculosis combined with Mycoplasma pneumoniae pneumonia. B. The discussion of the article is too lengthy, and the author should discuss the key issues, and the first and second paragraphs discussed in the current article should need to be refined.

Additional comments

To ensure the validity of the results, this study should follow the TRIPOD (Transparent Reporting Initiative for the Development and Validation of Clinical Predictive Models) checklist.The TRIPOD checklist contains a series of criteria and guidelines for assessing and enhancing the quality of clinical predictive models. These criteria include the type of model, sources of development and validation data, methods of feature selection and processing, model development process, performance evaluation, and clinical utility of the model. By following these meticulous guidelines, the transparency, reproducibility, and validity of research results can be ensured, thereby increasing their reliability and utility in clinical applications.

---

## Round 0.2 · Minor Revisions

The lasso is a regularization method, not a classifier or a regression method, but it can be used with classifiers and regression methods, having the power to perform embedded feature selection.

In this manuscript, the authors mention LASSO regression as an independent regression method, which is not right. Maybe the authors used LASSO logistics regression. If so, please delete logistics method.

**Language Note:** The review process has identified that the English language must be improved. PeerJ can provide language editing services - please contact us at copyediting@peerj.com for pricing (be sure to provide your manuscript number and title). Alternatively, you should make your own arrangements to improve the language quality and provide details in your response letter. – PeerJ Staff

Reviewer 1 ·

Basic reporting

The manuscript excels in clarity and precision of language, boasts a robust literature review grounding it in the field, and is well-structured with informative visuals and accessible raw data, effectively linking results to hypotheses.

Experimental design

The relevance of this study to the goals of the journal is clear, providing original research with clear objectives and replicable methods.

Validity of the findings

Reliable data and sound analysis are provided; however, assessment of impact and novelty is lacking, although the ample literature more appropriately compensates for this.

Reviewer 2 ·

Basic reporting

This work is distinguished by its articulate presentation, extensive and relevant literature references, meticulous structural and graphical clarity, and results that directly validate the research hypotheses.However, I believe that the author's language still needs a tremendous amount of revision and recommend that the author send the manuscript to a professional organization for polish-ups. Professional academic terminology should be accurate, which will help readers to read better.

Experimental design

The author's revisions go a long way toward refining the previous regrets, and I have no other doubts about the content of the study at this time.

Validity of the findings

As I described in the previous entry, the authors completed a very careful revision, after which the conclusions in the manuscript were well articulated, relevant to the original research question and limited to supporting the results.

Additional comments

I don't have any more comments.

---

## Round 0.3 · accepted · Accept

Authors have made revisions according to my comments. I think this paper can be accepted for publication,